# Markov state modelling reveals heterogeneous drug-inhibition mechanism of Calmodulin

**Annie M. Westerlund**[1☯]**, Akshay Sridhar**[1☯]**, Leo Dahl**[1]**, Alma Andersson**[1,2]**, Anna-Yaroslava Bodnar**[1]**, Lucie Delemotte**[1] *

**1** Department of Applied Physics, Science for Life Laboratory, KTH Royal Institute of Technology, Solna, Sweden, **2** Division of Gene Technology, Science for Life Laboratory, KTH Royal Institute of Technology, Stockholm, Sweden

☯ These authors contributed equally to this work.

* lucied@kth.se

**Data Availability Statement:** Code (Jupyter notebook) and input data needed to reproduce the results presented herein can be found on Zenodo with DOI 10.5281/zenodo.7045222.

## Abstract

Calmodulin (CaM) is a calcium sensor which binds and regulates a wide range of target-proteins. This implicitly enables the concentration of calcium to influence many downstream physiological responses, including muscle contraction, learning and depression. The antipsychotic drug trifluoperazine (TFP) is a known CaM inhibitor. By binding to various sites, TFP prevents CaM from associating to target-proteins. However, the molecular and state-dependent mechanisms behind CaM inhibition by drugs such as TFP are largely unknown. Here, we build a Markov state model (MSM) from adaptively sampled molecular dynamics simulations and reveal the structural and dynamical features behind the inhibitory mechanism of TFP-binding to the C-terminal domain of CaM. We specifically identify three major TFP binding-modes from the MSM macrostates, and distinguish their effect on CaM conformation by using a systematic analysis protocol based on biophysical descriptors and tools from machine learning. The results show that depending on the binding orientation, TFP effectively stabilizes features of the calcium-unbound CaM, either affecting the CaM hydrophobic binding pocket, the calcium binding sites or the secondary structure content in the bound domain. The conclusions drawn from this work may in the future serve to formulate a complete model of pharmacological modulation of CaM, which furthers our understanding of how these drugs affect signaling pathways as well as associated diseases.

## Author summary

Calmodulin (CaM) is a calcium-sensing protein which makes other proteins dependent on the surrounding calcium concentration by binding to these proteins. Such protein-protein interactions with CaM are vital for calcium to control many physiological pathways within the cell. The antipsychotic drug trifluoperazine (TFP) inhibits CaM's ability to bind and regulate other proteins. Here, we use molecular dynamics simulations together with Markov state modeling and machine learning to understand the structural

**Funding:** AMW was funded by a Swedish e-research center (SeRC) COVID-19 transition grant. AS was funded by Marie-Sklodowska Curie Fellowship Lipopeutics (Grant Number 898762). LDe acknowledges the Science for Life Laboratory (SciLifeLab), the Göran Gustafsson foundation and the Swedish Research Council (VR 2018-04905 and 2019-02433) for funding. The funders had no role in study design, data collection and analysis, decision to publish, or preparation of the manuscript.

**Competing interests:** The authors have declared that no competing interests exist.

and dynamical features by which TFP bound to the one domain of CaM prevents association to other proteins. We find that TFP encourages CaM to adopt a conformation that is like the one stabilized in absence of calcium: depending on the binding orientation of TFP, the drug indeed either affects the CaM hydrophobic binding pocket, the calcium binding sites or the secondary structure content in the domain. Understanding TFP binding is a first step towards designing better drugs targeting CaM.

## Introduction

Cellular $Ca^{2+}$-signaling pathways affect a multitude of physiological processes, including learning and memory, muscle contraction, metabolism, and long-term depression [1,2]. Calmodulin (CaM) is a small and highly-conserved $Ca^{2+}$-sensing protein which contributes to the ubiquity of $Ca^{2+}$-signaling by binding and regulating a wide range of target-proteins. Indeed, CaM has been found to regulate voltage-gated ion channels, G-Protein coupled receptors, NMDA receptors and even proteins with opposite cellular function like kinases and phosphatases [3,4].

CaM consists of N-terminal and C-terminal lobes (N-CaM and C-CaM), as well as a linker connecting the two lobes (Fig 1A). The binding promiscuity is partly due to the flexible linker, which permits a wrapping of the lobes around target-proteins, but also due to the conformational plasticity of the two lobes [3,5–7]. Each of the N-CaM and C-CaM lobes contains two

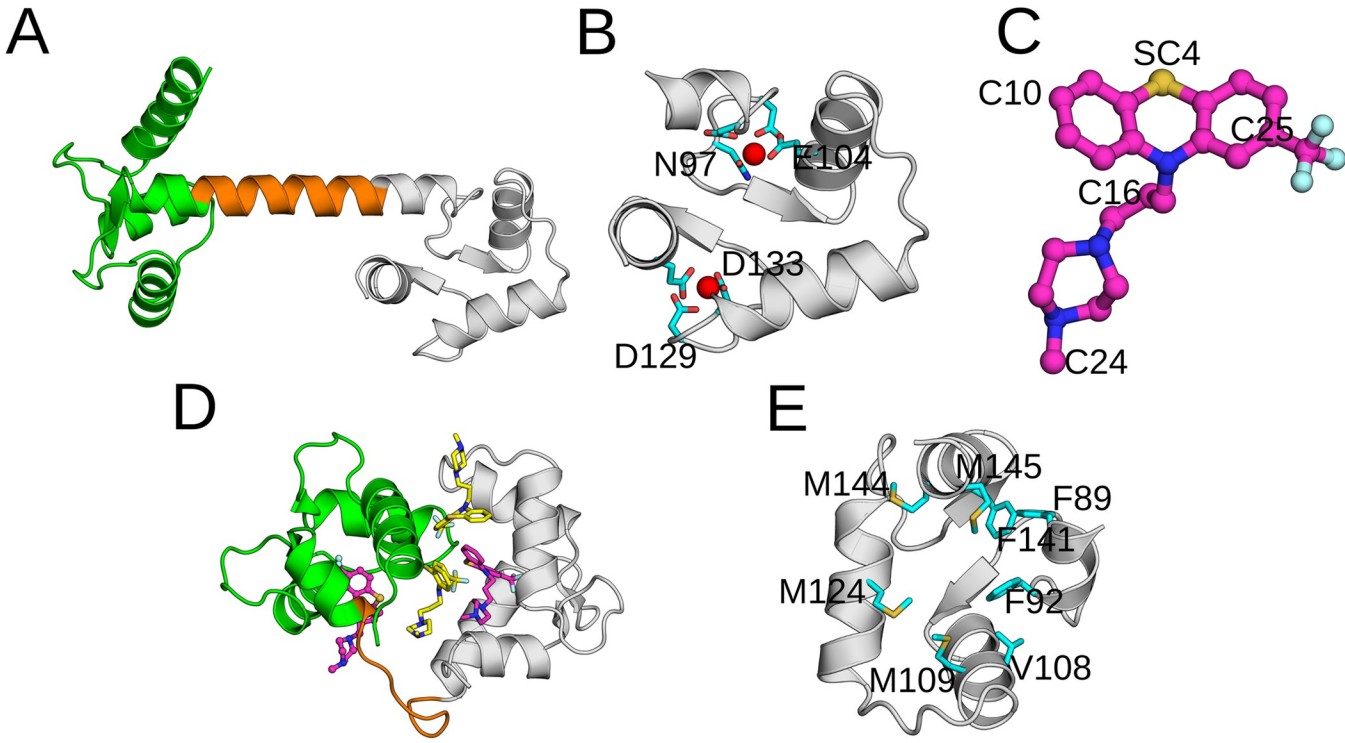

**Fig 1.** **(A)** The molecular structure of Calmodulin (PDB: 1CLL [8]). The N-CaM, linker and C-CaM are shown in green, orange and grey respectively. **(B)** The two EF-hand motifs making up the structure of C-CaM. The bound $Ca^{2+}$ ions are shown as red spheres together with their coordinating residues. **(C)** The molecular structure of Trifluoperazine. The five atoms distributed across the ligand used to featurize the trajectory are labelled. **(D)** Trifluoperazine binding configurations within calmodulin (PDB: 1LIN [23]). Ligands at the inter-lobe sites are coloured yellow and those at the N-CaM/C-CaM intra-lobe sites are coloured purple. **(E)** The hydrophobic methionine and aromatic residues making up the intra-lobe binding site within C-CaM.

(helix-loop-helix) EF-hand motifs (Fig 1B) which each form a $Ca^{2+}$ binding site [8–10]. The helices within each lobe enclose a hydrophobic pocket containing bulky aromatic residues and methionines that make up the interaction surface for target-proteins [11,12]. In line with the known binding promiscuity of CaM, various NMR, small-angle X-ray scattering (SAXS) and molecular dynamics (MD) simulation studies have demonstrated the protein's conformational flexibility [5,13–16]. Moreover, previous studies have demonstrated that $Ca^{2+}$-ion binding in the two lobes affects properties of the conformational landscape by inducing a change in the interhelical angles within the EF motifs [5,8,9,17]. Specifically, the $Ca^{2+}$-free (apo) CaM tends to adopt a closed conformation, shielding the hydrophobic residues from the surrounding solvent. $Ca^{2+}$-bound (holo) CaM, on the other hand, may adopt more open conformations where the target-protein interacting residues are exposed to solvent [5,13].

Antipsychotic phenothiazines make up a class of drugs which are known CaM modulators [18–21]. By binding to CaM, the drugs effectively inhibit target-protein association. Interestingly, the hydrophobicity of these drugs correlates with the ability to inhibit CaM function, suggesting important interactions with the hydrophobic binding pockets of the two lobes [19]. Trifluoperazine (TFP, Fig 1C) is a potent member of this family of drugs whose interaction with CaM is well-studied [22–25]. Apart from antipsychotic purposes, TFP has also been investigated as treatment for various cancer forms [20]. Crystallographic structures of TFP bound to CaM revealed that the drug is capable of binding to CaM with varying stoichiometries at both inter- and intra-lobe binding sites (Fig 1D).

TFP binding at the lower-affinity inter-lobe site is suggested to inhibit target-protein association by causing a compact CaM structure [26–28]. In contrast, the effect on CaM's ability to bind target-proteins due to intra-lobe drug binding is still poorly understood. In experimentally resolved CaM-TFP complex structures, TFP at this intra-lobe C-CaM site occupies two different poses where the drug interacts with the hydrophobic methionine and aromatic residues (Fig 1E) [25]. However, the mechanism and kinetics of transitions between the binding poses, as well as possible inhibitory effects on target-protein binding, remains unclear.

Molecular Dynamics (MD) simulations provide atomistic resolution of protein dynamic behavior and are routinely used to reveal the impact of drug-binding on protein function. However, conformational changes of CaM lobes, as well as phenothiazine binding, occur on the microsecond (μs) timescale; beyond the range of classical unbiased MD simulations [13,29]. Markov state modeling (MSM) is a computational technique that allows the stitching together of multiple short independent simulations and thus circumvents the sampling issue [30,31].

Here, we analyze the ensemble of TFP binding-modes in C-CaM, the most favorable binding site for TFP [32]. This is done by constructing an MSM from ~21 μs adaptively sampled atomistic simulations of CaM in presence of a bound TFP molecule. The obtained free energy landscape and kinetic network then validate the sampled drug-binding modes and illustrate their coarse structural and dynamical attributes. Through further systematic and comparative analysis of the MSM macrostates with previously sampled trajectories of TFP-unbound CaM, we uncover the state-dependent molecular determinants through which TFP prevents CaM from binding target-proteins.

## Materials and methods

### Calmodulin-TFP system preparation and equilibration

To enable extensive sampling of TFP bound to the multitude of C-CaM conformational states, we chose to initiate simulations in a diversity of conformational states. To do so, we obtained the three C-CaM states predicted from simulations of the 3CLN structure [33] in our prior

work [5]–states 2, 5 and 6, which correspond to Calmodulin's binding to partner proteins in shallow, intermediate and deep manner respectively (S1 Fig). The initial TFP positions within each state were generated by alignment of the C-CaM Cα to chain A of the 4RJD [25] structure. To account for the reduced ligand diffusivity within the collapsed hydrophobic core of state 2, we obtained an additional TFP initial pose by alignment of the C-CaM Cα to chain B of the 4RJD structure.

Each $Ca^{2+}$ bound CaM-TFP complex obtained thereby were placed in a box containing a solution of 22598 water molecules and $Na^+$ counter-ions. The system interaction energies were minimized until the maximum force on an atom was < 1000 kJ/mol and equilibrated by a 20 ps NVT ensemble with harmonic restraints of 1000 $kJ/mol/nm^2$ applied to protein and TFP atoms. Pressure was then relaxed, and the box scaled using a Berendsen barostat [34] for 20 ps with similar restraints on the protein and TFP atoms.

## Replica exchange solute tempering and adaptive sampling simulations

Before initiating the MD simulations for MSM construction, we ran replica exchange solute tempering (REST) simulations [35,36] to quickly explore the CaM-TFP conformational landscape and obtain a wide range of initial configurations to launch the initial round of adaptive sampling simulations. As such, the REST simulations were not run until convergence of the free energy landscape and not used in the MSM analysis. In temperature replica exchange (REMD) simulations, parallel simulations are run at different temperatures with the higher temperatures allowing a faster exploration of conformational states. REST simulations are a modification of the REMD algorithm wherein only a subset of the system is heated to enable a wider temperature span for the same number of replicas, thereby making the method more efficient for state exploration [36]. We initiated three sets of REST simulations from each of the three bound states with 5 replicas run for 500 ns each. Exchanges between adjacent replicas were attempted every 4 ps and the 'temperature' span (303.15 K to 330 K) was determined using the temperature predictor of Patriksson and van der Spoel [37] by only considering the CaM-TFP atoms. Efficiency of the REST simulations exchanges was assessed by analyzing the energy overlap between adjacent replicas and exchange probability (S2 Fig). Protein-ligand contacts of the lowest-temperature replica were then projected onto the two slowest degrees of freedom (called 'tICs'—see 'Selecting distance-based features and projecting the data onto slow degrees of freedom'). The two tICs were obtained from the continuous REST trajectories. For each CaM-TFP state, 10 points uniformly distributed across the two-dimensional grid spanning the extent of these two tICs were selected–and their representative conformations were used to initiate the first round of 100 ns-long unbiased MD simulations. Doing so ensured the initiation of MD simulations from both free energy basins and barriers with equal probabilities.

For subsequent rounds of adaptive sampling, we used a counts-based method to sample new initial configurations. This was done by first building an MSM (see 'Construction of a Markov state model') using the existing unbiased MD simulations and randomly selecting microstates to sample from, based on the stationary distribution over microstates. Specifically, the probability of sampling an initial configuration from a microstate was inversely proportional to its stationary probability [38]. Thus, microstates with a low stationary probability were more likely selected for seeding new simulations. In other words, rarely visited parts of the conformational landscape were more often chosen to launch new simulations. This simple counts-based method has been demonstrated to be efficient for exploring new states [38,39]. Through eight rounds of such adaptive sampling, we accumulated a combined simulation time of 20.8 μs across the initial C-CaM/TFP binding configuration.

## Molecular dynamics simulation parameters

The MD simulations were performed using Gromacs 2018 [40] with a timestep of 2 fs. For the REST simulations, we used Plumed 2.3.5 [41,42]. Long-range electrostatic interactions were calculated using the particle mesh Ewald method [43] and hydrogen-bond lengths were constrained using LINCS [44]. Pressure and temperature were maintained through the use of the Parrinello–Rahman barostat [45] (1 bar) and Nose-Hoover (300 K) thermostat [46] respectively. The protein was described using the Charmm36 force field [47], water using the TIP3P model, $Ca^{2+}$-ions using the Charmm27 parameters of Liao et al. [48] and TFP parameters were generated using STaGE [49].

## Selecting distance-based features and projecting the data onto slow degrees of freedom

To build the MSM, we first determined distance features to describe TFP binding-modes within C-CaM. For this, the minimum distances between the 60 C-CaM residues 88–147 and five TFP atoms (C25, C24, C10, SC4, C16) were calculated. The five atoms were chosen to encompass the ligand's functional groups (Fig 1C)–the C25 atom is within the aromatic ring while C10 is adjacent to the trifluoromethyl group, and the SC4 Sulphur is situated between the two. The C16 and C24 atoms, on the other hand, are situated within the alkyl linker and piperazine groups respectively. These distances were transformed into quasi-binary contacts, varying cut-offs from 5 to 8 Å, using the following transformations:

$$D = \begin{cases} D < i, \\ (D < i)^{-1} \end{cases} where\ i = 5, 6, 7, 8 \text{ Å})$$

To maximize the kinetic variance within the features, we evaluated these eight different feature types using the variational approach for Markov processes (VAMP2) score [50,51]. Each feature type was scored at 5 lag-times (2 ns, 5 ns, 10 ns, 15 ns, 20 ns) using the top ten eigenvalues and 5-fold cross-validation. S3 Fig shows the mean scores for different lag-times. The scores suggested that these feature types yielded MSMs of similar quality. However, the feature type given by a 6 Å cutoff and inverse distances yielded the highest mean VAMP2 score across lag-times, and this feature type was therefore selected for MSM construction.

The data was projected with time-lagged independent component analysis (tICA) [52] using time-lagged correlation matrices (tICA time-lag of 20 ns). A tICA projection provides a low-dimensional representation of the data along the slowest degrees of freedom. Overall, we used the python package MDTraj [53] to compute distance-based features, MSMBuilder [54] for parameter optimization and pyEMMA [55] for scoring the distance-based features, projecting the data, building the MSM and extracting the final macrostates.

## Construction of a Markov state model

The Generalized matrix Rayleigh quotient (GMRQ) method was used for MSM hyperparameter selection. S4 Fig shows the GMRQ scores calculated from 5-fold cross-validation, suggesting that the inclusion of the 10 slowest time-lagged independent components (tICs) and 200 microstates is sufficient to describe the Markov model. The 10 slowest time-lagged independent components (tICs) obtained from the full set of simulations were thus used as input to k-means clustering to obtain 200 microstates. To select a Markovian, or memoryless, lag-time for the MSM, we first estimated a set of transition probability matrices at varying lag-times. Each such matrix describes the probability of transitioning between microstates at a specific lag-time. By plotting the implied timescales [56,57] of the obtained transition matrices,

we identified a Markovian lag-time where the timescales appear to converge (15 ns–S5 Fig). The final MSM was constructed using this lag-time, and used to assign probability weights ($\pi_i$) to each trajectory frame i. Equilibrium averages of any system characteristic $O$ could then be obtained with a weighted sum,

$$\langle O \rangle = \frac{\sum \pi_i O_i}{\sum \pi_i}.$$

The number of MSM metastable states was selected using eigenvalue spectral analysis of the MSM transition matrix (S6 Fig). PCCA++ spectral clustering [58,59] was then used to estimate the probability of a trajectory frame belonging to each macrostate. Finally, we identified core-states by assigning a trajectory frame to one of the macrostates if the state probability was >80%. The rest of the trajectory frames were left as transition points. The core states allow us to better distinguish the states due to reduced noise in state-definition.

### Identification of important residues using machine learning

To pinpoint C-CaM residues participating in important residue-interactions of each identified TFP binding state, we used the *demystifying* toolkit [60]. This approach is based on explainable artificial intelligence (AI). In short, a machine learning model can be trained to recognize the three major states based on each frame's internal coordinates and metastable state-assignment. We may then ask the model which input features contributed the most to making a classification (state assignment) decision. Here, the input features were given by the inverse distances between all C-CaM residue pairs which were less than 6.5 Å in at least one frame and more than 6.5 Å in another. It should be noted that this contrasts with the feature types used for MSM construction which considered distances between CaM residues and selected TFP atoms. We trained random forest (RF) models using scikit-learn [61] and default parameters as determined in Fleetwood et al. [60] to recognize each metastable state given the input features and calculated the importance of residue-interactions for distinguishing between states by their Gini-importance [62]. Moreover, we calculated the Kullback-Leibler divergence of inverse residue-distance distributions for comparison. For both models, we trained one model per state in a one-versus-the-rest fashion. Finally, the per-residue importance was calculated as a sum over all the residues' interactions. The obtained residue importance was normalized between 0 and 1.

## Results and discussion

### Validation of Markov state model and identification of three major metastable states

We projected two-dimensional MSM free-energy landscapes along the three slowest time-dependent independent components (tICs). The best state separation was observed when the data was projected along tIC1-tIC3 (Figs 2A, S7 and S8). This free-energy landscape suggested the presence of three well-defined major local minima. The projection of the TFP-bound C-CaM structures onto the tIC space revealed how the free energy basins obtained using MD simulations compared with experimentally resolved structures: the 4RJD (Chain—or CaM molecule—B) and 1CTR structures indeed differ from 1A29 and 1LIN along tIC3. Interestingly, Chain A of the 4RJD structure is displaced from the other structures along tIC1, located close to a separate free-energy minimum. To further validate these free energy minima, we analyzed the paths taken by the various sampled trajectories and confirmed that we observe multiple transitions between the free-energy basins. Hence, the coverage of the conformational space by the combined set of MD simulations was extensive (S9 Fig).

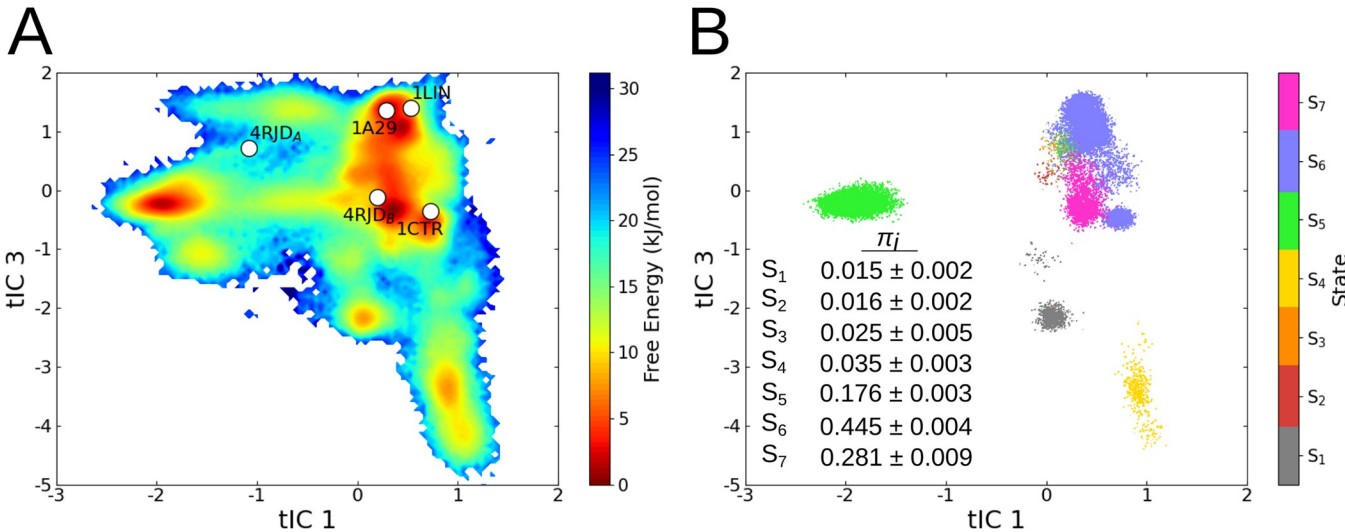

**Fig 2.** **(A)** Projection of the Markov state model free-energy surface along the time-lagged independent components tIC1 and tIC3. The projection of the 4RJD (Chains A & B) [25], 1LIN [23], 1A29 [24] and 1CTR [22] crystallographic structures onto the surface are shown as white dots. **(B)** $S_1$-$S_7$ macrostate assignments along the tICs obtained from PCCA++ clustering. Each trajectory frame is represented as a dot within the scatter plot. The stationary probability distributions of the seven macrostates and the associated standard deviations are shown in the inset.

Based on the MSM spectral analysis, we selected and extracted seven metastable states (S6 Fig). Fig 2B shows the resulting clustering, where each dot corresponds to a trajectory frame, colored by its macrostate assignment. The seven states ($S_1$-$S_7$), which were validated by Chapman-Kolmogorov tests (S10 Fig), are well separated in the tIC space (S7 and S8 Figs). We note, however, that the three major macrostates ($S_5$, $S_6$, $S_7$) accounted for ~98% of the stationary probability distribution (Fig 2B). We thus proceeded to characterize these three major TFP binding-modes and their effect on C-CaM conformations.

## The metastable states represent distinct TFP binding modes

We first identified the coarse molecular features of each metastable state by calculating the ligand root mean square displacement (RMSD) across frames in the state after alignment of the C-CaM Cα atoms. The trajectory frame with the lowest mean RMSD was then selected as a representative structure. A visual inspection of the resulting representative binding modes provides an initial explanation of their separation in tIC-space (Fig 3A). The trifluoromethyl ($CF_3$) moiety of TFP within $S_5$/$S_6$ and $S_7$, for example, is found in reversed orientations, either pointing towards or away from the hydrophobic pocket, respectively. Consistently, the experimentally resolved structures projected onto the free energy basins display similar 'flipped' configurations: the $CF_3$ group points towards the hydrophobic pocket in 4RJD (Chain A) [25], 1A29 [24] and 1LIN [23], and away from the pocket in 4RJD (Chain B) [25] and 1CTR [22].

Clearly, state $S_7$ represents a separate binding pose compared to $S_5$ and $S_6$. A distinct difference between $S_5$ and $S_6$ is, however, difficult to assess by visual inspection. Moreover, unlike the static representative structures, the ensembles of microstates provide a dynamic picture of residue movements and interactions with TFP. To account for the conformational heterogeneity within metastable core-states, we therefore analyzed the ensemble of frames assigned to each state. By doing so, we set out to reveal subtle, but important, differences in the interactions between TFP and CaM. First, we validated the pose difference across metastable states by studying the state-dependent interactions between TFP atoms and C-CaM residues. The

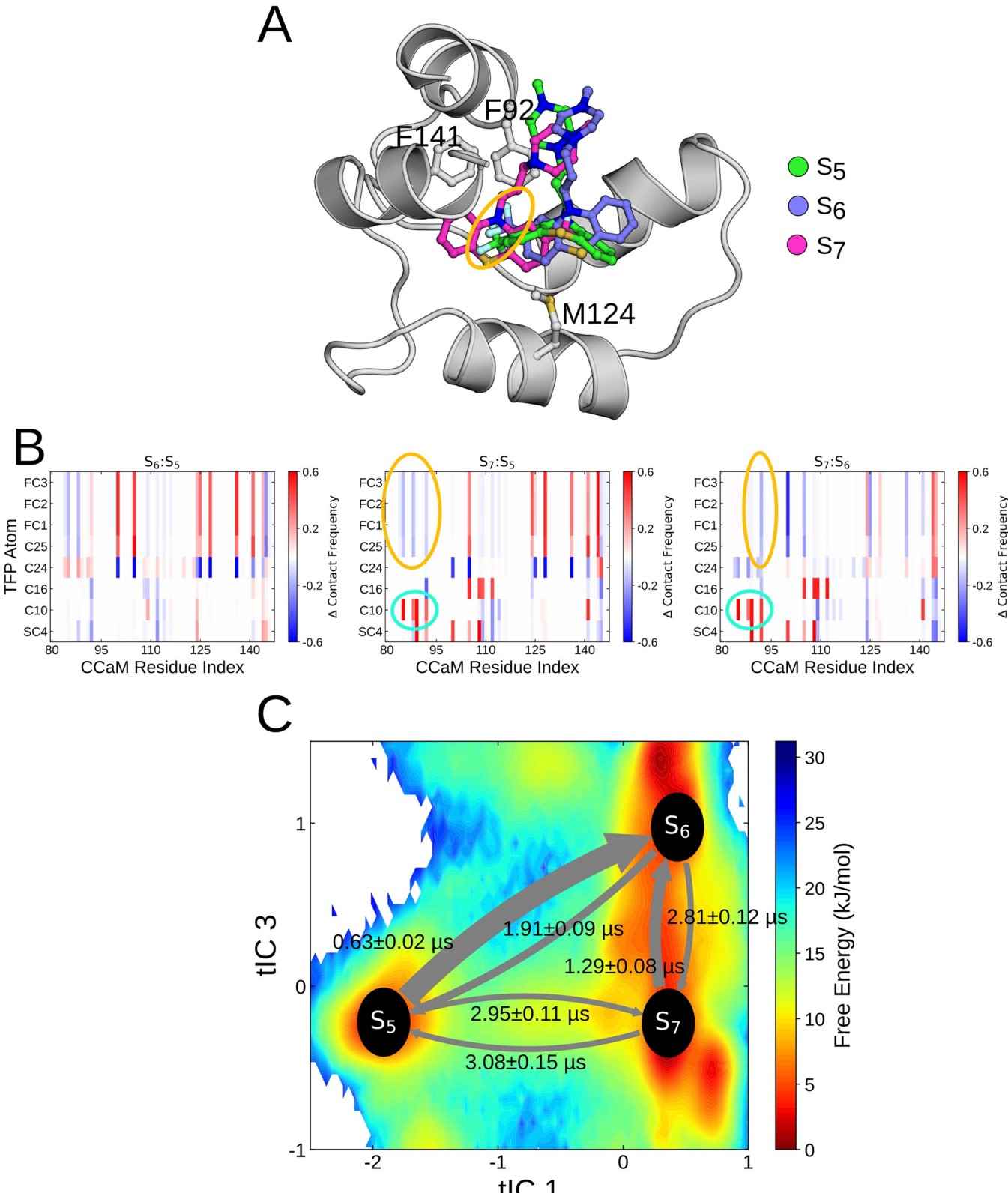

**Fig 3.** (**A**) Representative structures of the $S_5$, $S_6$ and $S_7$ macrostates aligned using the Cα C-CaM. The flipped orientation that buries the $CF_3$ moiety within the $S_5$ and $S_6$ macrostates is highlighted using an orange circle. The three F92, F141 and M124 residues whose interactions characterize *apo-* and *holo-*CaM states are illustrated. (**B**) Relative contact frequency of the TFP atoms with C-CaM residues compared between the three $S_5$, $S_6$ and $S_7$ macrostates. The flipped

orientation that buries the $CF_3$ moiety within the $S_5$ and $S_6$ macrostates is highlighted using an orange circle and the contrasting burying of the C10 atom with $S_7$ is highlighted using cyan circles. **(C)** Projection of the centres of the three populated macrostates onto the calculated free-energy surface along time-dependent independent components tIC1 and tIC3. Transition rates representing the mean first passage times between the macrostates and the associated standard deviations are illustrated as arrows.

difference in contact frequencies between the three metastable states is shown in Fig 3B. Consistent with a flipped orientation between $S_5$/$S_6$ and $S_7$, the C10 and C25 atoms located at opposite edges display similar delta contact frequencies within the $S_7$:$S_5$ and $S_7$:$S_6$ differences. In $S_5$ and $S_6$, the C25 atom and the Fluorines are buried to interact with the F89/F92 residues at the base of the hydrophobic pocket (orange circle). Alternatively, in $S_7$, the C10 atom slots into the hydrophobic pocket and interacts with the buried aromatic residues (cyan circle). The differences between the $S_5$ and $S_6$ binding poses, however, remained difficult to characterize. Indeed, these differences are subtle, and the orientation of the phenothiazine tricyclic is identical in both states. The atomistic contact analysis instead suggests that the main difference between the two binding-modes primarily stems from the interactions between C24 and the residues making up the binding pocket.

To depict the dynamics of interconversion between TFP binding modes, we calculated the mean first passage times (MFPT) [63] between the three major macrostates. The results demonstrate that the $S_5$ and $S_6$ states interchange on the sub-microsecond timescale while exchanges with the $S_7$ state occur on a significantly longer timescale, consistent with the structural differences observed above. Interestingly, the MFPTs also show that a transition to $S_7$ from the $S_5$ state with the buried TFP typically requires a longer timescale than a transition from the less buried TFP pose in state $S_6$. This hints that although $S_6$ is globally more similar to $S_5$, it may represent an intermediate state with subtle features similar to both $S_7$ and $S_5$.

## TFP binding-modes alter the C-CaM binding pocket by affecting local interactions at a $Ca^{2+}$-binding site

With the MSM, we were able to identify three major macrostates which agree with the experimentally resolved TFP-bound configurations. However, the backbone conformations of C-CaM in the representative structures are nearly identical, with a maximum Cα RMSD of only 1.71 Å between them. Hence, we hypothesize that the effect of TFP binding-modes on C-CaM conformation is instead manifested through subtle state-dependent changes of residue interactions. To pinpoint CaM residues with distinct interaction patterns in the three states, we utilized machine learning and explainable artificial intelligence [60]. Briefly, we used a random forest (RF) classifier to recognize the three major states based on internal coordinates and metastable state-assignment, and then extracted the per-residue accumulated feature importance (see 'Materials and Methods').

Fig 4A–4C show the state-dependent RF importance profiles. A high RF importance indicates large state-dependent changes in the residue's interaction pattern and thereby suggests movement of the residue relative to its interacting residues. Estimating importance from the Kullback-Leibler divergence between distances' probability distributions validates these importance profiles by yielding profiles which share main features with the ones obtained by RF (S11 Fig). As expected for $S_5$/$S_6$ and $S_7$ states with opposing orientations, the RF models identify M144 and F92 residues as important for discerning the states. This result is consistent with the C10 atom oriented towards these residues in the respective states [25]. In fact, both F92 and M144 are highly conserved residues [64], often interacting with target-proteins [7]. Earlier work showed that exposure of F92 to solvent favors deep binding to target-proteins [5], and that the F92A mutant is linked to a drop in ion affinity [65]. In addition to F92, we also

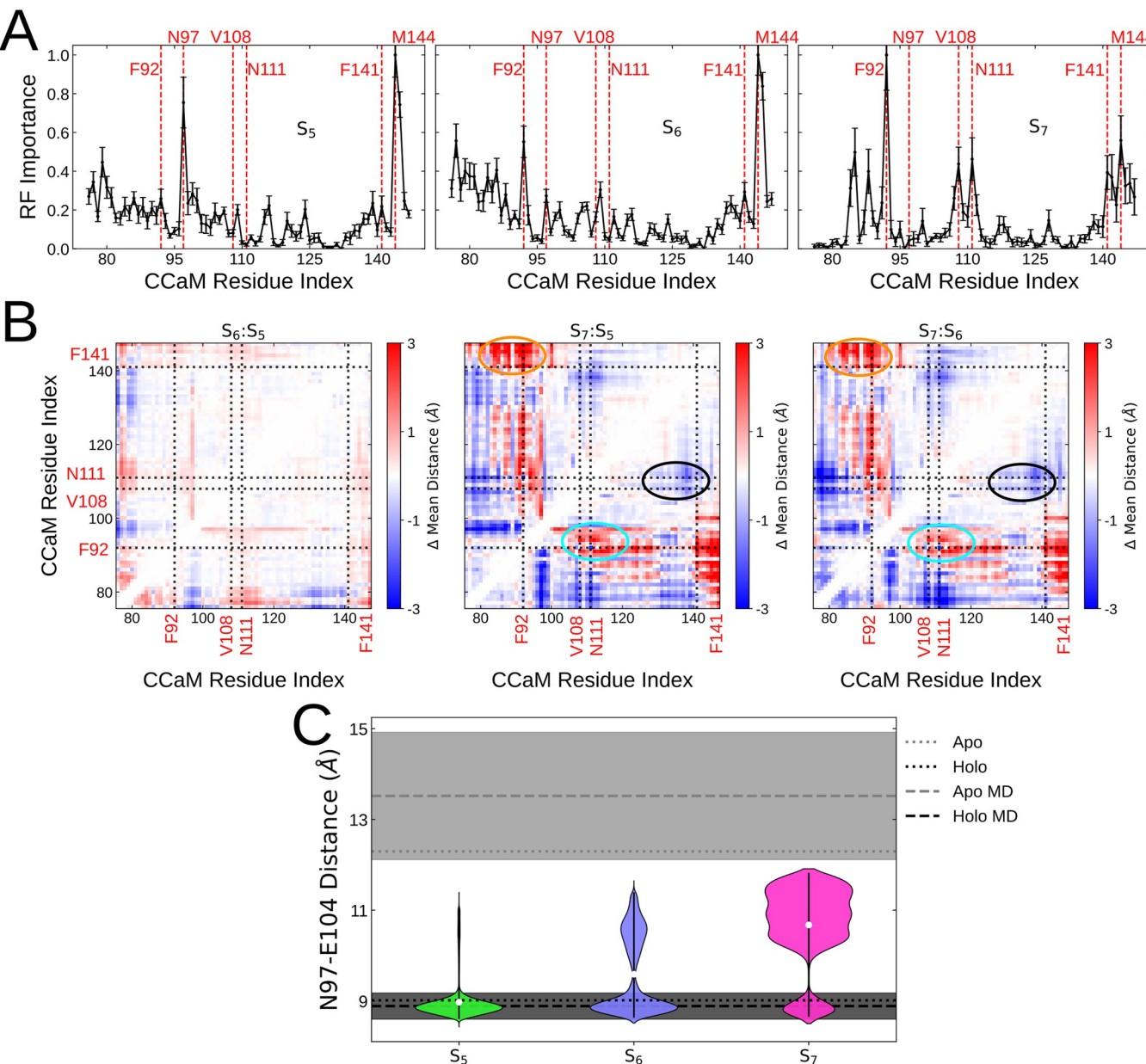

**Fig 4.** **(A)** The per-residue importance in discerning the $S_5$, $S_6$ and $S_7$ macrostates calculated using the supervised Random Forest method. Plots represent the mean importance values calculated from five-fold cross-validation and the standard deviations are plotted as error bars. Physiologically important residues and those with high importance values are illustrated using red dotted lines. **(B)** Relative C-CaM inter-residue distances compared between the $S_5$, $S_6$ and $S_7$ macrostates. The important F92, V108, N111 and F141 residues identified by the Random Forest method are illustrated as dotted lines. The increased distance between the F92-F141 and F92-V108 residue pairs in $S_7$ are highlighted as orange and cyan circles respectively. The decreased distance between the V108-F141 residues in $S_7$ is highlighted with a black circle/ **(C)** Distance between center-of-mass of the N97 and E104 residues making up the $Ca^{2+}$ binding site within the $S_5$, $S_6$ and $S_7$ macrostates. Each violinplot spans the 5th and 95th percentile of distances and densities are weighted by the Markov state model probabilities. The median value for each macrostate is represented as a white dot. The inter-residue distances calculated from *apo-* (PDB: 1CFD [9]) and *holo-* (PDB: 1CLL [8]) calmodulin structures are shown as grey and black dotted lines respectively. The mean inter-residue distances within MD simulations [5] of *apo-* and *holo-*CaM are shown as grey and black dashed lines respectively. The standard-deviations of the residue distances within the MD simulations are shown as a shaded area.

note that F141, another highly conserved residue [64], has a relatively high importance in the $S_7$ profile. Not only is F141L a known long QT syndrome (LQTS) mutation which also decreases $Ca^{2+}$-affinity [66,67], but the interaction and stacking of F141 and F92 has been

shown to associate with the transition from apo- to holo-like conformations [13]. Moreover, we observe an increased importance of V108 in $S_7$ (Fig 4A). This residue is indeed important for the packing and repacking of these aromatic residues during the transition from apo- to holo-like conformations [13]. It was also found important to characterize the overall conformational ensemble of C-CaM [60]. Finally, we note that N111 has a relatively high importance in the $S_7$ profile (Fig 4A). This residue has also been suggested to change interactions upon $Ca^{2+}$-binding [68]. To further rationalize the structural basis of the importance profiles, we calculated the change in mean inter-residue distances between the states (Fig 4B). Consistent with the RF importance predictions, distances between the helices comprising the F92 and F141 residues are further apart in $S_7$ than in $S_6$/$S_5$ (orange circle). The helix comprising the V108 and N111 residues show similar state-dependant differences with distance to the F92 helix farther in $S_7$ than in $S_6$/$S_5$ (cyan circle) and distance to the F141 helix farther in $S_6$/$S_5$ than in $S_7$ (black circle)–a result consistent with the residues' capability to impede hydrophobic stacking within the C-CaM core.

Next, we detect differences in the importance profiles of $S_5$ and $S_6$ (Fig 4A). We find that N97 is important for recognizing $S_5$, while F92 appears to change its interaction pattern in $S_6$. Interestingly, the N97I mutant has been shown to lead to a reorientation of the hydrophobic domain [3], while the N97S mutant causes LQTS [67]. The latter has specifically been shown to affect the activation of the voltage-gated ion channel KCNQ1 likely due to the interactions between CaM and the voltage-sensor domain during channel activation [69,70]. The estimated importance of F92 is likely due to its position within the hydrophobic pocket and interaction with TFP (Fig 3A). N97, on the other hand, is on a loop distant from the hydrophobic pocket, and instead coordinates one of the bound $Ca^{2+}$ ions (Fig 1B). To assess the state-dependence of residue interactions within this $Ca^{2+}$-binding site, we computed the distribution of distances between the N97 and its interacting residue E104 for each state (Fig 4C). In the $S_5$ state, the residues are stable at a spatially close distance, a characteristic of holo CaM. The flipped TFP pose in state $S_7$, however, disrupts the ion-binding site and leads to a more *apo*-like conformation with a larger separation of the residues. Incidentally, TFP binding in C-CaM has been suggested to markedly reduce the $Ca^{2+}$ affinity [32], possibly due to such disruptions at the ion binding site. Consistent with the higher affinity of the second $Ca^{2+}$-binding site [71], however, the TFP binding modes display no noticeable disruption on its stability (D129-D133 –S12 Fig).

## Flipping of the TFP molecule affects the hydrophobic pocket and is associated with a changed β-sheet structure content

As mentioned above, the process of binding and unbinding $Ca^{2+}$ is intrinsically associated with CaM conformational changes. In the presence of ions, the lobe undergoes a 'repacking' of hydrophobic residues which is characterized by the movement of the F92 residue. This repacking results in a stacking of four aromatic residues (S13 Fig) [13]. Motivated by the findings from the state-dependent RF importance profiles, we therefore investigated how the different TFP binding-modes affect the hydrophobic pocket. To characterize this, we studied the interactions between F92, M124 and F141 –three residues from different helices whose sidechains subtend into the pocket (Fig 1E). In fact, M124, together with M109, M144 and M145, make up the set of highly conserved methionines [64] which often participate in target-protein binding [11,12].

Fig 5A shows the hydrophobic packing along two inter-residue distances–the F92-F141 and F92-M124 distances. In contrast to the other two states, the slotting of the C10 atom with its aromatic ring in $S_7$ prevents the otherwise advantageous stacking interactions of F92 with the

other aromatic residues (Fig 3A). This leads to the adoption of a conformation reminiscent of the crystallographic apo-CaM (Fig 5A). However, the observed binding mode-dependent effects are local. As such, the packing of the methionines lining the pocket is left unaffected (S14 Fig). Viewing the $S_7$ as a state with local apo-like structural features is consistent with the findings that the transition from apo- to holo-like conformation is mainly characterized by the repacking of aromatic residues rather than changed interactions between the methionines [13,72]. In summary, our results suggest that the F92, F141 and N97 residues play particularly significant roles in the coupling between the hydrophobic domain and $Ca^{2+}$-site, and that the orientation of TFP controls the switch between these apo- and holo-like interactions.

The two $Ca^{2+}$ sites in C-CaM are cooperative with ion-binding at one interface resulting in up to a 10 kJ/mol enhancement in binding at the other interface [5,6]. Moreover, binding $Ca^{2+}$ at the two sites yields a shift in the position of the β-sheet structure within the lobe [5]. To investigate whether the state-dependent TFP-binding can also affect the apo- and holo-like hallmarks of secondary structure content, we compared the per-residue secondary structure frequency in each state with that of the drug-unbound CaM ensemble (Fig 5B) [5].

The $_{98}$GYISA$_{102}$ and $_{134}$GQVNY$_{138}$ loops are adjacent to the two ion-sites and transiently form antiparallel β-sheet structures [73]. Specifically, the Y99 and N137 residues participate more frequently in sheet structures in the $S_5$ and $S_6$ states (Fig 5B and 5C). Recent work has shown that Y99 and N137 are important for the functional interactions between CaM and the activated-open state of KCNQ1 [70]. Together, the results provide a hypothesis on the mechanism of Calmodulin inhibition by the varying TFP binding modes. Apart from obstructing the hydrophobic pocket, in the $S_5$ and $S_6$ states, TFP could potentially hinder interactions between CaM and target-proteins by locking the crucial Y99 and N137 residues in a β-sheet structure. Conversely, within the $S_7$ state, TFP functions by enabling a local switch to an *apo*-like state characterized by differential interactions of the aromatic F89, F92 and F141 residues.

## Conclusions

The rationalizing of protein-ligand interactions forms the basis of drug design. The structural heterogeneity of CaM, which forms the basis of its function, however, makes it a challenging drug target [74]. The inhibition of CaM requires a drug to bind across its conformational ensemble, including transitioning between *apo-* and *holo-like* states. To understand the molecular aspects of such an inhibition, we performed extensive unbiased and adaptively sampled MD simulations of TFP-bound (holo) CaM to build an MSM from which metastable TFP-binding modes could be extracted. The results demonstrate that TFP is a heterogeneous inhibitor which acts by blocking the different CaM binding pockets via various binding poses. The burial of the halogen moiety into the hydrophobic pocket, for example, correlates with C-CaM adopting subtle *holo*-like features around a $Ca^{2+}$ binding site, while the 'flipped' TFP-binding mode could be attributed to apo-like features in this region. Moreover, we observed a general apo-like secondary structure content in C-CaM due to TFP binding. This, together with the observed state-dependent packing of hydrophobic residues in C-CaM, hints that TFP may prevent target-protein binding through subtle yet distinct blocking mechanisms. The different binding-poses may affect the coupling between the C-CaM β-sheet structure, a $Ca^{2+}$-binding site and the packing of aromatic residues.

An extension of the current work may address the construction of MSM models for understanding TFP binding to N-CaM. Since N-CaM is less dynamic than C-CaM [16], a comparison can shed further light on the inhibitory mechanism of TFP. Nonetheless, the notion that TFP uses different binding poses to block CaM is in line with the conformational heterogeneity and binding-promiscuity of CaM. This work thus deepens our understanding of how one

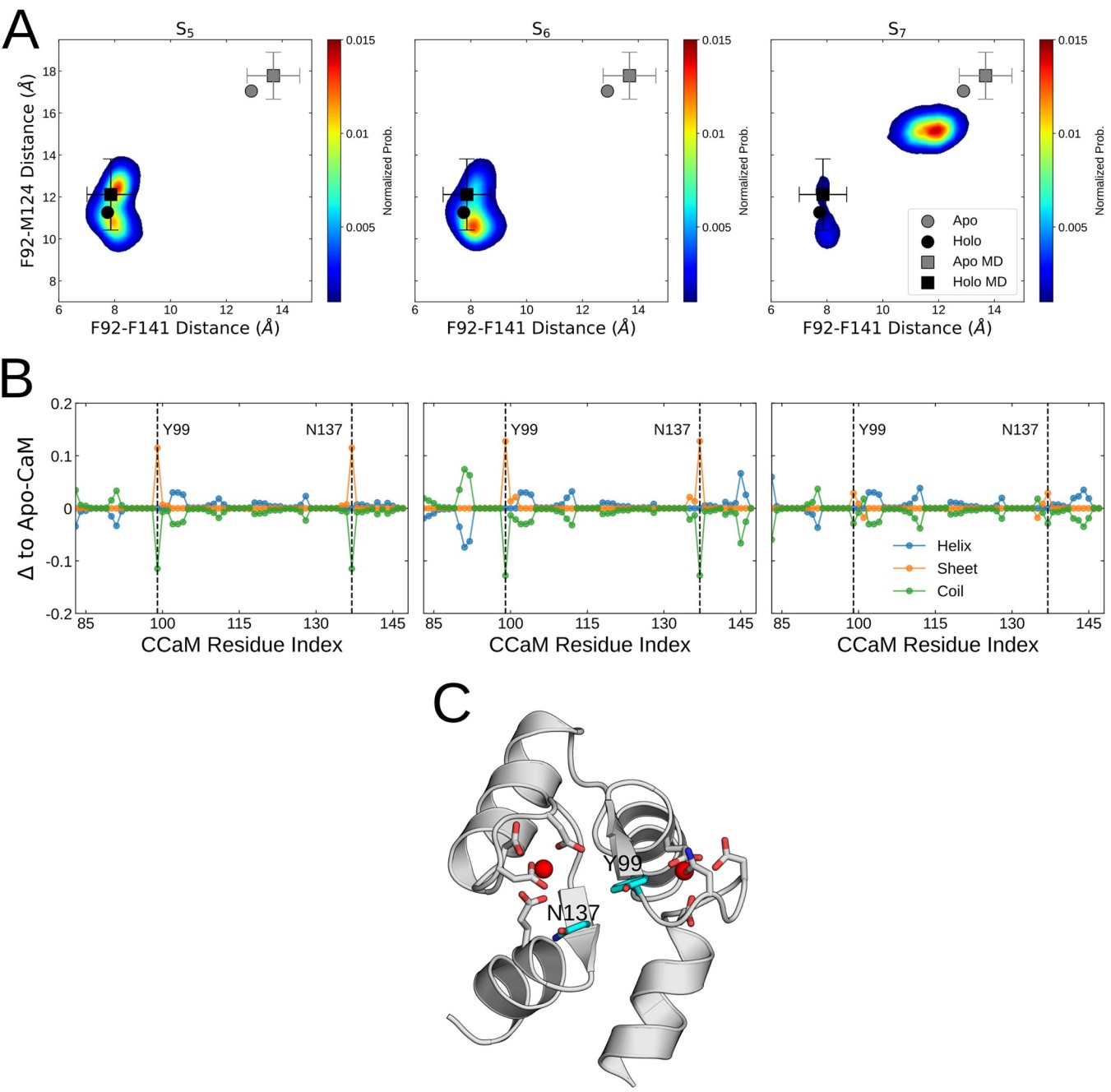

**Fig 5. (A)** Inter-residue distances between the sidechains of the F92, M124 and F141 within the $S_5$, $S_6$ and $S_7$ macrostates. The two-dimensional histograms are weighted by their Markov state model probabilities. The inter-residue distances calculated from *apo-* (PDB: 1CFD [9]) and *holo-* (PDB: 1CLL [8]) calmodulin structures are shown as grey and black dots respectively. The mean inter-residue distances within MD simulations [5] of *apo-* and *holo-*CaM are shown as grey and black squares respectively. Error bars represent the standard deviations of residue distances within the MD simulations. **(B)** The per-residue secondary structure of C-CaM within the three populated macrostates compared to their behaviour within simulations of *apo-*CaM. The Y99 and N137 residues displaying enhanced sheet behaviour are illustrated as dotted lines. **(C)** The molecular structure of C-CaM illustrating the Y99-N137 antiparallel sheet situated adjacent to the two Ca$^{2+}$ ion binding sites. The bound Ca$^{2+}$ ions are shown as red spheres.

drug can inhibit such highly flexible binding pockets. The results presented here may serve as a stepping-stone towards a full understanding of the pharmacological modulation of CaM, with implications on signaling pathways and associated diseases.

## Supporting information

**S1 Fig. Calmodulin states 2, 5 and 6 obtained from Molecular Dynamics simulations of the 3CLN structure that were used as initial configurations for the REST simulations.** The bound $Ca^{2+}$ ions included in the simulations are shown as red spheres.
(TIFF)

**S2 Fig. Efficiency of the Replica-Exchange with Solute Tempering (REST) simulations initiated from the three CaM states assessed by the energy overlap and mean exchange acceptance probability between adjacent replicas.**
(TIFF)

**S3 Fig. VAMP2 scores of the 10 slowest processes for different feature transformations calculated at a variety of lag times τ.** The mean values from five-fold cross-validation are plotted as bars and error bars represent the standard deviations. The mean value across the lag times calculated for each feature transformation is mentioned within the legend.
(TIFF)

**S4 Fig. The optimization of MSM hyperparameters through the calculation of GMRQ scores for different feature transformations at varying numbers of microstates and processes.** The mean values from five-fold cross-validation are plotted as dots and standard deviations are plotted as error bars.
(TIFF)

**S5 Fig. Top 8 eigenvalues of the transition probability matrix calculated at varying lagtimes to identify a memoryless Markovian time.** The 95% confidence intervals of the eigenvalues are shown as shaded regions. The black solid curve delimits a shaded region where the implied timescales are shorter than the lagtime.
(TIFF)

**S6 Fig. Spectral analysis of the eigenvalues to identify the number of clusters.** The cutoff between the sixth and seventh relaxation timescales selected in this work is illustrated as red dotted line.
(TIFF)

**S7 Fig. Top: Projection of the Markov state model free-energy surface along the three slowest time-lagged independent compo- nents (tICs).** Bottom: S1-S7 macrostate assignments along the three tICs obtained from PCCA++ clustering. Each trajectory frame is represented as a dot within the scatter plot.
(TIFF)

**S8 Fig. Separation of the S5, S6 and S7 macrostates along the 10 slowest time-lagged independent components (tICs) used for Markov state model construction.**
(TIFF)

**S9 Fig. Transitions between the macrostate basins analyzed by projecting the initial and final configurations of individual trajectories onto the free-energy surface.** Individual trajectories are represented as subplots with the initial and final configuration shown as black and white dots respectively. Trajectories with transitions between the basins are highlighted with a magenta outline.
(TIFF)

**S10 Fig. Chapman-Kolmogorov test validating the Markov state model by comparing the probabilities of transiting between the macrostates (blue line) and the calculated**

**probabilities from the constructed model (black line).**
(TIFF)

**S11 Fig. The per-residue importance in discerning the S5, S6 and S7 macrostates calculated using the supervised KL Divergence method.** Plots represent the mean values calculated from five-fold cross-validation and the standard deviations are plotted as error bars. Physiologically important residues and those with high importance values are illustrated using red dotted lines.
(TIFF)

**S12 Fig. Distance between center-of-mass of the D129 and D133 acidic residues making up the second $Ca^{2+}$ binding site within the S5, S6 and S7 macrostates.** Each violinplot spans the 5th and 95th percentile of distances and is weighted by the Markov state model probabilities. The median value for each macrostate is represented as a white dot. The inter-residue distances calculated from apo- (PDB: 1CFD) and holo- (PDB: 1CLL) calmodulin structures are shown as grey and black dotted lines respectively.
(TIFF)

**S13 Fig.** Transition in the stacking of aromatic residues between the (A) apo- (PDB: 1CFD) and (B) holo- (PDB: 1CLL) states of calmodulin induced by the binding of Ca2+ ions.
(TIFF)

**S14 Fig. Distances between the sidechains of the M109, M124 and M144 Methionine residues making up the hydrophobic binding pocket within the S5, S6 and S7 macrostates.** Each violinplot spans the 5th and 95th percentile of distances and is weighted by the Markov state model probabilities. The median value for each macrostate is represented as a white dot. The inter-residue distances calculated from apo- (PDB: 1CFD) and holo- (PDB: 1CLL) calmodulin structures are shown as grey and black dotted lines respectively.
(TIFF)

## Acknowledgments

The MD simulations were performed on resources provided by the Swedish National Infrastructure for Computing (SNIC) on Beskow at the PDC Center for High Performance Computing (PDC-HPC). We further wish to thank Magnus Lundborg for assisting in ligand parameterization.

## Author Contributions

**Conceptualization:** Annie M. Westerlund, Akshay Sridhar, Lucie Delemotte.

**Data curation:** Annie M. Westerlund, Akshay Sridhar, Leo Dahl, Alma Andersson, Anna-Yaroslava Bodnar.

**Formal analysis:** Annie M. Westerlund, Leo Dahl, Alma Andersson, Anna-Yaroslava Bodnar.

**Funding acquisition:** Lucie Delemotte.

**Investigation:** Annie M. Westerlund, Akshay Sridhar, Leo Dahl, Alma Andersson, Anna-Yaroslava Bodnar, Lucie Delemotte.

**Methodology:** Annie M. Westerlund, Akshay Sridhar.

**Project administration:** Annie M. Westerlund, Akshay Sridhar, Lucie Delemotte.

**Resources:** Lucie Delemotte.

**Software:** Akshay Sridhar.

**Supervision:** Annie M. Westerlund, Lucie Delemotte.

**Validation:** Annie M. Westerlund, Akshay Sridhar, Leo Dahl, Alma Andersson, Anna-Yaroslava Bodnar, Lucie Delemotte.

**Visualization:** Annie M. Westerlund, Akshay Sridhar, Anna-Yaroslava Bodnar.

**Writing – original draft:** Annie M. Westerlund, Akshay Sridhar.

**Writing – review & editing:** Annie M. Westerlund, Akshay Sridhar, Leo Dahl, Alma Andersson, Anna-Yaroslava Bodnar, Lucie Delemotte.

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
