## [Decision Letter · Decision Letter 0]

12 Jul 2022

Dear Dr Delemotte,

Thank you very much for submitting your manuscript "Markov State Modeling Reveals Heterogeneous Drug-Inhibition Mechanism of Calmodulin" for consideration at PLOS Computational Biology.

As with all papers reviewed by the journal, your manuscript was reviewed by members of the editorial board and by several independent reviewers. In light of the reviews (below this email), we would like to invite the resubmission of a significantly-revised version that takes into account the reviewers' comments.

We cannot make any decision about publication until we have seen the revised manuscript and your response to the reviewers' comments. Your revised manuscript is also likely to be sent to reviewers for further evaluation.

Sincerely,

Guanghong Wei

Associate Editor

PLOS Computational Biology

Nir Ben-Tal

Deputy Editor

PLOS Computational Biology

Reviewer's Responses to Questions

**Comments to the Authors:**

Reviewer #1: The authors combined molecular dynamics simulations, markov state model and machine learning strategies to investigate the structural dynamics of CaM exerted by the binding of one known drug TFP, which prevents the downstream protein associations. The authors claimed that the TFP binding to CaM can put an influence on its C-terminal region and induce it into a Ca2+-free conformation. This current manuscript provides some structural insights into the regulatory roles of TFP in stabilizing CaM. Despite that, many technical flaws should be fixed and more structural analyses should be provided in the revised version before the final acceptance. See below:

1. To validate the MSM, I would suggest the authors to perform the GMRQ analyses for the hyperparameters selections of the MSM, including the tICA lag-time, microstate number, number of tICs etc. Related to this, the implied timescale curves shown in Fig. S2 are actually not well leveled off, I suggest the authors to further increase the lag-time to check if the curves can converge or try alternative models.

2. In Fig. 2 and Fig. 3, the errors of the stationary distribution and MFPTs are not provided, this should be fixed in the revised manuscript.

3. From Fig. 1D, TFP can potentially bind with CaM in different pockets and varied TFP/CaM ratios. The authors should provide more details regarding the system setups. For example, how many TFP molecules were included in the system? How to parametrize the TFP molecule in the study etc. One structural model for the complete system is recommended.

4. In Fig. 4, the authors highlighted several critical residues in CaM, however, no detailed structural analyses to support their conclusions. I would suggest the authors to conduct more structural analyses and discuss with their ML results.

5. The authors attempted to compare the TFP-bound CaM with its Ca2+-bound/unbound state, however, either Ca2+-bound or -unbound CaM has been simulated in current work. I would suggest the authors to perform several additional MD simulations for these systems in order to make a valid comparison.

Reviewer #2: Overall, the authors detail the various mechanisms of TFP interactions with Calmodulin (CaM) by performing molecular dynamics simulations and adaptively sampling the conformational landscape of the CaM-TFP complex. The authors aim to explain the mechanism of TFP inhibition, and use extensive molecular dynamics simulations for the same. The authors use a guided machine-learning and quantitative approaches to explain the binding modes of TFP to CaM, which gives further validity to the results. They aim to explain the hydrophobic rearrangements of the binding pocket are responsible for the inhibition of CaM. It is an interesting work and I have a long list of minor edits. Further review is not needed.

Introduction:

Paragraph 4: Inconsistent terminology is used – intra-lobe is referred to as within-lobe and vice-versa.

Paragraph 5: ‘MD Simulations may be used?’ Authors could change this line to “MD simulations are routinely used” ..

Methods:

Calmodulin-TFP system preparation and equilibration:

Was the N-Terminal Domain present in the simulations, but the TFP only bound to the C-Terminal domain? The manuscript does not make this clear.

The authors say ‘TFP was placed in prominent states’ – can they make the said ‘placement’ clear? What are states 2, 5 and 6 from prior work? Can the authors justify their choice? From how many total states were these three chosen? Were these simulations in the presence of calcium ions? Please make the methodology more clear.

Replica Exchange solute tempering and adaptive sampling simulations:

The authors perform Replica Exchange Simulated Tempering (REST) simulations. However, there is not enough motivation provided for the rationale behind these simulations. Why are the authors performing these simulations? A mention of the method in the introduction with some relevant background is needed.

The conformations were chosen uniformly over the grid – what was the criteria for ensuring uniformity?

Could the authors provide more details into the data collected per round of sampling?

Selecting distance-based features and projecting the data onto slow degrees of freedom

“The minimum distances between all C-CaM residues and five TFP atoms distributed across the ligand’s structure (C25, C24, C10, S, C16) were calculated.” Why these 5 atoms in particular are chosen for TFP?

Is the S in (C25, C24, C10, S, C16) same as SC4 in Fig 1C? If yes, it is used inconsistently.

Fig S1: What do the values in the legend 0.5, 0.6 indicate? Can the authors make the term ‘feature type’ clear?

Construction of a Markov State Model

‘Results were stable’ – can the authors make it more quantitative justification than a qualitative one?

Fig S2: Have the authors computed the errors for the implied timescales?

Identification of important residues using machine learning

The cutoffs for the MSM construction (6Å) differ from the cutoff used for identification (6.5Å) – can the authors justify this decision?

Results and discussion:

Validation of Markov state model

The Chapman-Kolmogorov state test differs for the S2-S2 conversion diverges significantly between the predicted and the estimated probabilities.

Why tIC1 and tIC3 were chosen as the slowest metrics ? Why not tIC2 ?

The metastable states represent different binding modes

The slowest process corresponding to tIC1 – shows significant separation among macrostates S5 and S6 in Fig 2B, but these two macrostates represent similar binding modes. Why?

Authors mention an orange circle and cyan circle, in Fig 3 but it seems to be missing.

Is there evidence of the TFP interconverting through the different binding modes?

TFP Binding modes alter the C-CaM binding pocket by affecting local interactions at a Ca+2 binding site

The C_α RMSD of 1.71Å is unclear – is it between S5-S6, S6-S7 or S5-S7?

Can the authors give a more qualitative explanation of the y-axis in Fig 4A – what does RF importance signify? How does RF importance correlate to the dynamics of the protein macrostates S(5-7)?

Flipping of the TFP molecule affects the hydrophobic pocket and is associated with a changed beta-sheet structure content

Have the authors performed simulations in the presence of Ca+2 to explore the reduced affinity binding between Ca+2 in the presence of TFP?

Reviewer #3: In this manuscript, the authors have constructed Markov State Models (MSMs) from molecular dynamics (MD) simulations to study the binding of trifluoperazine to calmodulin. They identified important protein residues that are associated with several drug binding poses. Interestingly, they find that upon binding, trifluoperazine can stabilize both apo and holo-like calmodulin conformations, depending on the binding pose. Overall, this is a rigorous study, and the manuscript is also well-written. Their results provide new insights into the understanding of the calmodulin inhibition and may facilitate the drug development in the long term. Therefore, I would like to recommend its publications after minor revision (see my comments below):

1. To obtain the input features for the tICA analysis, the authors simply chose all the pair-wise distances between protein residues and five TFP atoms within a cut-off (6\\AA selected by the VAMP2 score). I am wondering if the application of Spectral-OASIS, SRVs or other methods (see discussions in JACS Au, 1(9), 1330, (2021)) can help further refine the input feature set? More interestingly, will these methods (Spectral-OASIS, SRVs, etc) can identify the same set of important residues (e.g., distance features between these important residues and TFP) as those obtained from their explainable AI algorithm after the MSM construction?

2. The current discussion of binding poses is very detailed in terms of the importance of protein residues but lacking in terms of their inhibitory effect. Specifically, it is not clear how the binding pose inducing the holo-like conformation contributes to the inhibition of calmodulin. I would like to suggest the authors to expand their discussions of the binding poses with respect to the subsequent inhibiting effect to benefit a general audience.

3. It will be helpful to include a SI figure to display the efficiency and convergence of their REST simulations (e.g., acceptance probability, coverage of the replica space, etc).

4. Fig S1: it is not obvious to me that “the feature type given by a 6 \\AA cutoff and inverse distances yielded the highest mean score across lag-times”. Could the authors clarify on this point?

5. The choice of hyperparameters, including tICA lag time, number of tICs, and number of microstates can all impact the quality of an MSM. These hyperparameters can also be optimized using the cross-validation method based on the VAMP2 score or GMRQ. The authors may consider optimizing some of these parameters or include some discussions on this point. Especially on the choice of number of microstates, could the authors clarify on their criterion: “until the results described herein were stable”?

6. Fig. S2: it’s not obvious to me that the implied timescales are fully converged. These implied timescale plots still display noticeable deviations from being flat even on the logarithm scale. Can the authors rephrase their claim of convergence and explain their choice of a relatively short MSM lag-time of 15ns? I notice that their 7-macrostate MSM (Fig. S6) is well validated by the Chapman-Kolmogorov test. Maybe adding the Chapman-Kolmogorov test of the 200-microstate MSM (on the residence probabilities of the top populated microstates) can help justify their choice of the Markovian lag time?

**Have the authors made all data and (if applicable) computational code underlying the findings in their manuscript fully available?**

Reviewer #1: None

Reviewer #2: **No: **It will be a few GB dataset after removing water so authors could consider sharing the entire MD dataset.

Reviewer #3: Yes

PLOS authors have the option to publish the peer review history of their article (what does this mean?). If published, this will include your full peer review and any attached files.

Reviewer #1: No

Reviewer #2: No

Reviewer #3: No
---

## [Decision Letter · Decision Letter 1]

18 Sep 2022

Dear Dr Delemotte,

We are pleased to inform you that your manuscript 'Markov State Modelling Reveals Heterogeneous Drug-Inhibition Mechanism of Calmodulin' has been provisionally accepted for publication in PLOS Computational Biology.

Best regards,

Guanghong Wei

Academic Editor

PLOS Computational Biology

Nir Ben-Tal

Section Editor

PLOS Computational Biology

Reviewer's Responses to Questions

**Comments to the Authors:**

Reviewer #1: The authors have addressed all of my raised issues.

Reviewer #2: Authors have addressed all the minor comments outlined in the initial review. Congratulations to the authors on this interesting work.

Reviewer #3: The authors have appropriately addressed my comments in the revised manuscript. I would like to recommend its publication.

**Have the authors made all data and (if applicable) computational code underlying the findings in their manuscript fully available?**

Reviewer #1: None

Reviewer #2: Yes

Reviewer #3: Yes

PLOS authors have the option to publish the peer review history of their article (what does this mean?). If published, this will include your full peer review and any attached files.

Reviewer #1: No

Reviewer #2: **Yes: **Diwakar Shukla

Reviewer #3: No

---

## [Editor Report · Acceptance letter]

2 Oct 2022

PCOMPBIOL-D-22-00880R1 

Markov State Modelling Reveals Heterogeneous Drug-Inhibition Mechanism of Calmodulin

Dear Dr Delemotte,

I am pleased to inform you that your manuscript has been formally accepted for publication in PLOS Computational Biology. Your manuscript is now with our production department and you will be notified of the publication date in due course.

With kind regards,

Zsofia Freund
